# PRIVATE AND DEBIASED MODEL TRAINING: A FAIR DIFFERENTIAL PRIVACY GRADIENT FRAMEWORK

## ABSTRACT

Deep learning models are vulnerable to leak private information about the training data. Differential privacy (DP) is increasingly implemented in deep learning to preserve the data privacy through different ways, one of which is imposing DP to the gradients in training models, called DP gradients. Unfortunately, adding DP to gradients has negative impacts on either robustness or fairness, and even both of deep learning models, resulting in unexpected performance of their data management tasks. In this paper, we undertake deep exploration of the disparate impact of DP gradients and their mitigating. Specifically, through empirical analysis we disclose that gradient variance renders clear disparate impact on different groups, and provide the theoretical proof on the relations between gradient variance and model fairness. Then we develop a Fair Differential Privacy Gradient (FDPG) framework to alleviate the disparate impact of DP gradients while protecting the data privacy. To implement the novel framework, we create a fairness-aware sampling mechanism to restore balance among groups, and design the adaptive noise injection strategy to maintain model utility. Our experimental evaluations demonstrate the effectiveness of FDPG on multiple mainstream classification tasks in both single and multiple protected group attributes.

## 1 INTRODUCTION

Deep learning has gained significant attention for its remarkable advancements in abundant applications (He et al., 2016; Garcia et al., 2025). The success of deep learning models depends on a substantial amount of high-quality data. However, these data often contain sensitive personal information, which can be leaked by the model parameters (Fredrikson et al., 2015; Liu et al., 2022). For example, the Google Prediction API and Amazon Machine Learning have the potential to leak membership information from purchase records (Ye et al., 2022). Hence, training or (and) publishing private models is a prospective approach to defend data privacy leakages.

Differential Privacy (DP) (Dwork et al., 2006) is a provable and quantifiable method for privacy protection, and has been increasingly employed to prevent leakage of private training data in deep learning (Bu et al., 2024; Wagner, 2025; Zhao et al., 2025b). Currently, imposing DP in deep learning primarily involves the output parameter, objective function, and gradient (Jayaraman & Evans, 2019; Chen et al., 2023), in which the first one is on the publishing model stage, and the latter two during the training model stage. Imposing DP to gradients, namely DP gradients, is a more popular way to apply DP in deep learning (Wu et al., 2019; Chen et al., 2023).

Unfortunately, DP gradients can introduce or exacerbate bias of deep learning models, where accuracy drops much more for the disadvantaged classes or groups (known as disparate impact), undermining the fairness of the decision-making process and reiterating possible discrimination (Bagdasaryan et al., 2019; Farrand et al., 2020; Zhao et al., 2025a)). Thus, integrating both privacy and fairness in DP gradients is necessary and expected to deploy trustworthy deep learning models.

In these regards, we aim to seek answers to the following questions:

*Is there impact of gradient variance on the fairness of models with DP gradients? What is the relationship between gradient variance and the model fairness? How can we develop a framework to mitigate unfairness while maintaining model utility in DP gradients?*

To sum up, we address the posed questions through the following contributions:

- We provide the experimental disclosures regarding obvious relations between gradient variance and fairness in DP gradients, and present theoretical proof that gradient variance can be utilized to reduce disparate impact, thus enhancing model fairness.

- We propose a novel Fair Differential Privacy Gradient (FDPG) framework for private and debaised model training so as to mitigate the disparate impact while maintaining model utility.

- To implement the framework, we design a fairness-aware sampling mechanism by reweighting group samples in a privacy-preserving way and develop an adaptive noise injection strategy to control the influence of noise.

- Our extensive experiments encompass both single and multiple protected group attributes, showing that FDPG can effectively mitigate disparate impact with satisfactory performance across diverse datasets in private model training.

## 2 RELATED WORK

**DP Gradients:** Differential privacy (DP) gives rigorous privacy guarantees about what can be inferred from the output of an algorithm (Jagielski et al., 2019). Existing DP gradient methods can be divided into two categories (Yang et al., 2022; Xiao et al., 2023) based on how to bound the sensitivity. The first is to clip the gradients with a constant threshold before injecting noise, i.e., differentially private stochastic gradient descent (DP-SGD) (Abadi et al., 2016), which is the first to introduce gradient clipping to ensure bounded sensitivity on gradients, then add noise to the clipped gradients. Subsequent works (Andrew et al., 2021; Papernot et al., 2021; Fu et al., 2024; Sha et al., 2024; Zhang et al., 2025; Wei et al., 2025) mainly focus on further improving the privacy-utility tradeoff in DP-SGD. The second is to normalize the gradients before noise injection, i.e., AUTO-S (Bu et al., 2024), which performs gradient normalization to retain information about the relative size of the gradients and get rid of the dependence of tuning on the clipping threshold (Liu et al., 2023), eventually achieving state-of-the-art performance (Bu et al., 2024). Unfortunately, these methods introduce disparate impact, thereby reducing fairness when applied to private model training (Bagdasaryan et al., 2019; Farrand et al., 2020).

**DP Gradients and Fairness:** Fairness is treating individuals or groups without any prejudice based on their inherent or acquired characteristics (Mehrabi et al., 2021). Achieving fairness in deep learning models is well-studied (Cao et al., 2024), recent works focus on both fairness and privacy in deep learning (Fioretto et al., 2022; Tran et al., 2021; Chang & Shokri, 2021; Lyu et al., 2022; Zhao et al., 2025a; Demelius, 2025). However, Chang & Shokri (2021) empirically found that imposing fairness constraints on private models could lead to higher loss for certain groups. Hence regular fairness-aware strategies cannot be directly applied to private models. To achieve the goal that the cost of adding privacy to a non-private model must be fairly distributed between groups, i.e., to alleviate the disparate impact, existing works (Xu et al., 2021; Tran et al., 2021; Esipova et al., 2023; Zhao et al., 2025a; Kim et al., 2025) attribute it to factors such as clipping errors, gradient norms, and gradient misalignment, and propose adaptive clipping or alignment-based methods for mitigation. However, they did not consider the multiple protected group attributes, and their solutions are tailored for models trained with DP-SGD without considering the DP Gradient methods using gradient normalization, such as AUTO-S. Although directly applying existing mitigation method (Kulynych et al., 2022) on models trained with AUTO-S can reduce unfairness to some extent, it will cause serious performance degradation. Hence, it is necessary to effectively mitigate the disparate impact while maintaining model utility in DP gradients.

## 3 PRELIMINARY

This section introduces some related concept of DP. Let $\|\cdot\|$ denote the $\ell_2$ norm of a vector and $\langle\cdot,\cdot\rangle$ denote the inner product of two vectors. We consider $f(\mathbf{x})$ as the loss of model $\mathbf{x}$. The gradient of $f(\mathbf{x})$ is represented by $\nabla f(\mathbf{x})$. Given a dataset $D$ with $n$ samples, $D_k$ denotes a subset of $D$ with the set of samples with protected group attribute $S = k$, where $k$ refers to a protected group, and there are $K$ groups in $D$. $C$ is clipping threshold, $\sigma$ is noise multiplier, $q_k \in (0, 1]$ represents the group probabilities with $\sum_{k \in S} q_k = 1$.

DP (Dwork et al., 2006) is a strong privacy notion used to ensure that the output data of the algorithm does not significantly change when a sample is changed.

**Definition 1 (Differential Privacy (Dwork et al., 2006))** *A randomized mechanism $\mathcal{M} : \mathcal{D} \to \mathbb{R}$ with domain $\mathcal{D}$ and range $\mathbb{R}$ is $(\epsilon, \delta)$-DP if for all pairs of datasets $D$ and $D'$ that differ by the addition or removal of one sample (neighboring datasets), and for any subset of outputs $\mathcal{O} \in \mathbb{R}$, $\Pr(\mathcal{M}(D) \in \mathcal{O}) \leq \exp(\epsilon) \cdot \Pr(\mathcal{M}(D') \in \mathcal{O}) + \delta$, where $\epsilon$ is the privacy budget, $\delta$ is a broken probability that the property does not hold. Smaller values of $\epsilon$ and $\delta$ indicate stronger privacy guarantee.*

By adding random noise, we can achieve differential privacy for a function $f : \mathcal{D} \to \mathbb{R}$. The sensitivity of $f$ determines how much noise is needed and is defined as follows.

**Definition 2 (Sensitivity (Dwork et al., 2006))** *Given function $f : \mathcal{D} \to \mathbb{R}$, with a differentially private mechanism is via additive noise calibrated to $f$'s sensitivity $S_f$, which is defined as $S_f = \max_{D,D'} \|f(D) - f(D')\|$, where $D$ and $D'$ are neighbor datasets.*

Existing DP gradient methods can be divided into two categories according to the way of bounding the sensitivity of the gradients. One way to bound the sensitivity of the gradients is gradient clipping, which is adopted by DP-SGD (Abadi et al., 2016), we describe gradient clipping as $\bar{g}_i = \mathbf{Clip}(g_i; C) = g_i \times \min\left(1, \frac{C}{\|g_i\|}\right)$, where $g_i = \nabla f_{\xi_i}(\mathbf{x})$ is the gradient of $i$-th random sample $\xi$. An alternative is to force a bounded sensitivity through gradient normalization, which is applied by Auto-S (Bu et al., 2024), gradient normalization is described as $\bar{g}_i = \mathbf{Normalize}(g_i; \gamma) = \frac{g_i}{\|g_i\| + \gamma}$, where regularization term $\gamma \in (0, 1]$.

Gradient variance is a measure of the difference between the average gradient of the entire dataset and the stochastic gradient during model training (Wang & He, 2021). Like most works (Gorbunov et al., 2020; Koloskova et al., 2023; Sadiev et al., 2023), it is assumed that gradient variance is bounded. The bounded variance is defined as follows.

**Definition 3 (Bounded Variance (Koloskova et al., 2023))** *For a randomly selected sample $\xi$ from dataset $D$ and all $\mathbf{x} \in \mathbb{R}^d$, the gradient variance is bounded by $\tau^2$, i.e.,*

$$\mathbb{E}_{\xi \in D} \left[\|\nabla f_\xi(\mathbf{x}) - \nabla f(\mathbf{x})\|^2\right] \leq \tau^2. \tag{1}$$

# 4 DISPARATE IMPACT OF DP GRADIENTS

In this section, we explore the correlation between fairness and gradient variance in DP gradients, including two representative DP gradient methods DP-SGD (with gradient clipping) (Abadi et al., 2016) and AUTO-S (with gradient normalization) (Bu et al., 2024). Disparate impact is measured in private model training by the disparity in performance degradation of different groups. The experimental setup of this empirical analysis can be found in Section 6.

## 4.1 EMPIRICAL ANALYSIS OF DISPARATE IMPACT

**The disparate impact of DP Gradients.** Fig. 1 illustrates the accuracy on CelebA dataset at models trained with DP-SGD and AUTO-S. We can observe a notable decrease in the accuracy of male in DP-SGD and AUTO-S compared to stochastic gradient descent (SGD), while the accuracy for female remains high, indicating that DP gradients result in disparate impact for groups. Especially in DP-SGD, the disparate impact becomes more severe with smaller $C$. Furthermore, the average loss for different groups in Adult, Dutch, MNIST and CelebA are presented in Fig. 2. The results reveal that models with DP gradients perform significantly better for the advantaged group (Female in Adult, Male in Dutch, class 2 in MNIST and Female in Celeb A) than for the disadvantaged groups, where disadvantaged groups experience a more pronounced increase in loss than their advantaged counterparts compared with SGD. DP gradients intensify the discrepancy in loss among different groups, deteriorating the fairness compared with SGD.

**The Correlation between Fairness and Gradient Variance.** In order to characterize the effect of DP gradients across different groups, we introduce the concept of gradient variance, which represents the dispersion of gradients (Wang & He, 2021). Fig. 2 depicts the results of average loss, and gradient variance for each group (i.e., the mean gradient variance across all samples in the group) across multiple models and entire datasets. Detailed experimental settings are provided in App. C.3. It is evident that there is a positive correlation between group loss and gradient variance, i.e., the model consistently demonstrates higher loss on

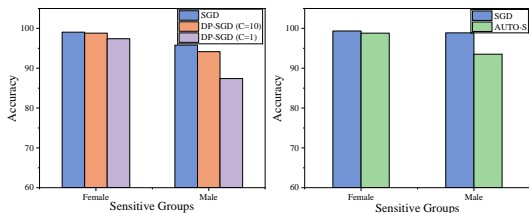

Figure 1: Accuracy of each group in the CelebA dataset. The privacy budget is $\epsilon = 2.5$ and $\epsilon = 8$ respectively.

a group with a larger gradient variance. In addition, model with DP gradients does not result in excessive increase of loss in groups with smaller gradient variance. For instance, DP-SGD has a more significant increase in loss and a larger gradient variance on class 8 than class 2 compared with SGD in MNIST.

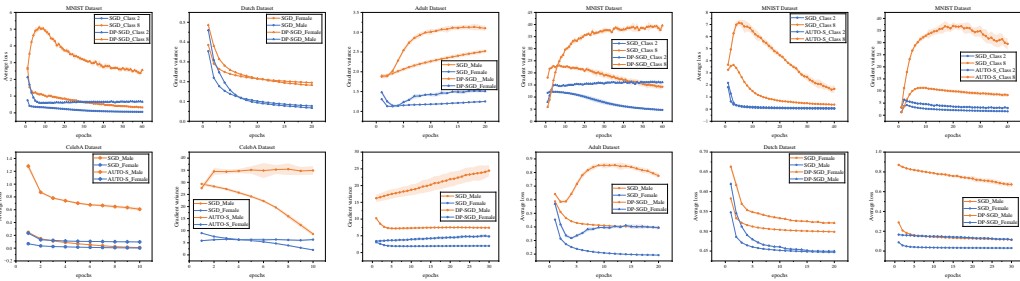

Figure 2: Measurements on model's loss and gradient variance per group.

In summary, disadvantaged groups generally exhibit larger gradient variance compared to those of advantaged groups. Thus, models with DP gradients suffer more severe performance degradation on disadvantaged groups. Next we provide the theoretical proof on above empirical observations of relations between fairness and gradient variance.

## 4.2 THEORETICAL ANALYSIS OF DISPARATE IMPACT IN DP-SGD

DP-SGD consists of gradient clipping and noise injection before averaging the gradients over the batch in iteration $t$ (Xiao et al., 2023), i.e. the algorithm can be defined as $\mathbf{x}_{t+1} = \mathbf{x}_t - \eta \left( \frac{1}{b} \sum_{i \in B_t} g_i \times min \left(1, \frac{C}{\|g_i\|}\right) + \mathbf{z}_t \right)$, where $\mathbf{z}_t \sim \mathcal{N}\left(0, \sigma^2 C^2 \mathbb{I}\right)$ is the Gaussian noise, $b$ is batch size, $\eta$ is learning rate, $B_t$ denotes the set of samples used in the $t$-th iteration.

According to (Xu et al., 2021), the loss discrepancy caused by DP-SGD can be measured by the error between the original and the DP gradients. We denote the gradient before clipping over the group $k$ ($k \in S$) as $G^k = \frac{1}{b^k} \sum_{i=1}^{b^k} g_i{}^k$, where $g_i^k$ represents the gradient of a random sample from group $k$ with sample size $b^k$. The gradient after clipping as $\bar{G}^k = \frac{1}{b^k} \sum_{i=1}^{b^k} \bar{g}_i^k$, and the gradient after clipping and noise as $\tilde{G}^k = \frac{1}{b^k}(\sum_{i=1}^{b^k} \bar{g}_i^k + \mathbf{z}_t)$, then we have the error due to DP-SGD by applying the Minkowski inequality (Kuczma, 2009):

$$\mathbb{E}|\tilde{G}^k - G^k| \leq \mathbb{E}|\tilde{G}^k - \bar{G}^k| + \mathbb{E}|\bar{G}^k - G^k| \leq \frac{1}{b^k}C^2\sigma^2 + \frac{1}{b^k}\sum_i^{b^k} \max(0, \|g_i^k\| - C)$$

$$= \frac{1}{b^k}C^2\sigma^2 + \frac{1}{b^k}\sum_i^{m_k}(\|g_i^k\| - C), \tag{2}$$

where $m^k$ is the number of samples that get clipped in group $k$. Eq. 2 consists of two terms: noise error $\frac{1}{b^k}C^2\sigma^2$ and clipping error $\frac{1}{b^k}\sum_i^{m_k}(\|g_i^k\| - C)$. For noise error, if $C$ is small, the noise error will be close to zero. For clipping error, it could lead to discrimination for each group, which means that given the clipping threshold $C$, the clipping error in the group with large gradient norms is larger than the one in the group with small gradient norms. Hence, gradient clipping has disparate impact on the gradient for each group, which in turn affects the model accuracy. Inspired by the convergence result of DP-SGD from (Koloskova et al., 2023), we can explain this observation through Theorem 1 which verifies that gradient variance directly influences the upper bound of the gradient norm.

**Theorem 1 (Convergence Guarantee (Koloskova et al., 2023))** *If $f$ is $(L_0, L_1)$-smooth (but not necessarily convex) and we run DP-SGD for $T$ iterations with step size $\eta \leq 1/[9(L_0 + CL_1)]$, then the gradient norm is upper bounded by $\mathcal{O}\left(\frac{L\eta}{C}\sigma^2 + \sqrt{L\eta}\sigma + \min\left(\tau^2, \frac{\tau^4}{C^2}\right) + \eta L\frac{\tau^2}{b} + \frac{F_0}{\eta T} + \frac{F_0^2}{\eta^2 T^2 C^2}\right)$, where $L = L_0 + \max_t \|\nabla f(\mathbf{x}_t)\| L_1$ and $F_0 = f(\mathbf{x}_0) - f^*$, $f^* = f(\mathbf{x}_*)$, and $\mathbf{x}_* = arg\,min_{\mathbf{x}}f(\mathbf{x})$.*

**Remark 1** *This theorem implies that, set aside the Gaussian noise with variance $\sigma$, provided the step size is small enough, DP-SGD suffers from a deviation term $\min\left(\tau^2, \frac{\tau^4}{C^2}\right)$, this term is controlled by gradient variance and clipping threshold, which means that by reducing the gradient variance and clipping threshold, this term can be decreased, further helping to produce smaller gradients and move towards the optimum, and eventually reducing the clipping error illustrated in Eq. 2.*

According to Theorem 1, we can infer that a group with a larger gradient variance corresponds to a larger $\tau$, which in turn implies larger gradient norms, ultimately leading to a larger clipping error. This is in harmony with the discoveries in Section 4.1. Hence, we can reduce the gradient variance of disadvantaged groups to balance the utility loss of each group.

### 4.3 THEORETICAL ANALYSIS OF DISPARATE IMPACT IN AUTO-S

Different from DP-SGD, AUTO-S is updated within mini-batch as follows (Yang et al., 2022):

$$\mathbf{x}_{t+1} = \mathbf{x}_t - \eta\left(\frac{1}{b}\sum_{i \in B_t}\frac{g_i}{\|g_i\| + \gamma} + \mathbf{z}_t\right), \tag{3}$$

where $\mathbf{z}_t \sim \mathcal{N}\left(0, \sigma^2\mathbb{I}\right)$. From Eq. 2, we note that noise has no impact on fairness. Hence we only consider the normalization error here. We leverage $G^k$ and $\bar{G}^k$ to denote the gradient of group $k$ before and after normalization respectively. Calculating the error due to normalization, we have:

$$\mathbb{E}|\bar{G}^k - G^k| = \frac{1}{b^k}\sum_i^{b^k}|\frac{g_i^k}{\|g_i^k\| + \gamma} - g_i^k| = \frac{1}{b^k}\sum_i^{b^k}|\frac{(1-\gamma)\|g_i^k\| - \|g_i^k\|^2}{\|g_i^k\| + \gamma}| \tag{4}$$

For simplicity, we denote $F(a)$ as $F(a) = \left|\frac{(1-\gamma)a - a^2}{a+\gamma}\right|$, where $a = \|g_i^k\|$, let $F(a) = 0$, then we have: $F(a) = \left|\frac{(1-\gamma)a - a^2}{a+\gamma}\right| = \left|\frac{(1-\gamma-a)a}{a+\gamma}\right| = 0$. Hence, $a = 0$ or $a = 1 - \gamma$. We observe that $F(a)$ has an upper bound when $a \leq 1 - \gamma$ (see Appendix D.1 for the visualization). Taking the derivative of $F(a)$ and let it be zero, we have: $\nabla F(a) = |\frac{\gamma - (a+\gamma)^2}{(a+\gamma)^2}| = 0$. Then $\gamma - (a + \gamma)^2 = 0$, we have $a = \sqrt{\gamma} - \gamma$, finally the upper bound is $F(\sqrt{\gamma} - \gamma) = 1 - 2\sqrt{\gamma} + \gamma$.

Note that the normalization error is also related to the gradient norm from Eq. 4. As shown in Fig. 5, the analysis of normalization error needs to be divided into two cases:

1. When $\|g_i^k\| \leq 1 - \gamma$, the normalization error has an upper bound $1 + \gamma - 2\sqrt{\gamma}$, which is only determined by the constant $\gamma$;

2. When $\|g_i^k\| > 1 - \gamma$, the normalization error intensified with the gradient norm $\|g_i^k\|$.

In the first case, normalization error has no impact on fairness. However, the second case implies that the normalization error in the group with large gradient norms is larger than the one with small gradient norms. Hence, we present Theorem 2 to validate that the gradient variance also has a direct impact on the upper bound of the gradient norm in AUTO-S. The proof of Theorem 2 is in Appendix A.2.

**Theorem 2** *Under bounded variance assumption, if $f$ is $(L_0, L_1)$-smooth, for $T$ iterations of Eq. 3 with step size $\eta$, then the gradient norm is upper bounded by $\mathcal{O}\left(\tau + \frac{F_0}{\eta T}\right)$.*

**Remark 2** *According to Eq. 1, the gradient variance is bounded by $\tau^2$, and the final result demonstrates that AUTO-S introduces a persistent deviation (determined by gradient variance) to the optimization process that cannot be eliminated by increasing the number of iterations, which means we can reduce the gradient norm by decreasing the gradient variance.*

Similar to the case of DP-SGD, we can reduce the disparate impact by balancing the gradient variance for each group in AUTO-S.

## 5 MITIGATING THE DISPARATE IMPACT

To mitigate the disparate impact in DP gradients, we propose Fair Differential Privacy Gradient (FDPG) framework (Fig. 3), which consists of a fairness-aware sampling mechanism and an adaptive noise injection strategy. We describe FDPG in Algorithm 1 (in the APP. B), where the fairness-aware sampling mechanism is designed to mitigate disparate impact by balancing the gradient variance among groups. On the other hand, the adaptive noise injection strategy can maintain model utility by controlling the injected noise. For simplicity, we only adopt the clipping method to describe our

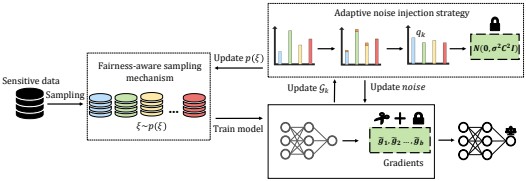

Figure 3: Overview of our Fair Differential Privacy Gradient (FDPG) framework.

framework, but all the following analysis can be generalized to the normalization operation with $C = 1$.

### 5.1 FAIRNESS-AWARE SAMPLING MECHANISM

Existing common data sampling methods are not designed to improve model fairness, hence we create a novel Fairness-aware Sampling mechanism to replace the original uniform sampling scheme to reduce disparate impact. We focus on balancing the gradient variance of each group. The intuition is to raise the gradient contribution of the disadvantaged groups in batch selection. The mechanism has two steps. We first need to incorporate fairness into the sampling probability for batch selection, then protect the intermediate statistics that contain sensitive information with DP.

**Considerations on the fairness.** From our theoretical analysis, bias is reflected in the relationship with gradient variance. Hence, an ideal operation is to reweight the samples in each group with gradient variance. The sampling probability for group $k$ can be defined as:

$$q_k = \frac{Var_k}{\sum_{i=1}^{K} Var_i}, \tag{5}$$

where $Var_k$ is the gradient variance for group $k$. Note that Eq. 5 involves computing the gradient variance during each training batch, which is a computationally expensive process. Based on the previously established relationship (Theorem 1 and Theorem 2), we can use the gradient norm instead of the gradient variance to compute group probabilities (i.e., the sampling probabilities for different groups) while reducing computational cost, we make two meaningful adjustments on Eq. 5. Firstly, we sample data from different groups with probability proportional to the average norm of gradients that get clipped/normalized, we have $\mathcal{G}_k = \frac{\widehat{G}_k}{m_k} = \frac{\sum_{i \in D_k \cap \|g_i\| > C} \|g_i\|}{|\{i: \|g_i^k\| > C\}|}$, where $D_k$ denotes

the samples from group $k$, $\widehat{G}_k$ is the sum of clipped/normalized gradient norm for group $k$, and $m_k$ is the number of samples that get clipped/normalized in group $k$. Secondly, in order to avoid redundant computation of per-sample gradients, we leverage the gradients from previous epoch to instruct training for the next epoch. The probability for group $k$ can be defined as $q_k = \frac{\mathcal{G}_k}{\sum_{i=1}^{K} \mathcal{G}_k}$.

**Protections on the statistical information.** To obtain group probabilities for fairness-aware sampling in a differentially private way, we need to release two additional values derived from protected information in each epoch: $\widehat{G}_k$ and $m_k$. Hence, we inject Gaussian noise into these two statistics to ensure DP. For $\widehat{G}_k$, since the sensitivity of actual gradients is hard or even impossible to estimate, we utilize a auxiliary bound $\beta$ to scale down the original per-sample gradient norm to preserve the magnitude information of gradients, we then perturb $\widehat{G}_k$ with Gaussian noise as $(\sum \min(\frac{\|g_i\|}{\beta}, 1) + N(0, \sigma_2^2 \mathbb{I})) \cdot \beta$, where $i \in D_k \cap \|g_i\| > C$, $\sigma_2$ represents the variance of Gaussian noise, and $min(\cdot)$ makes sure gradient norm has sensitivity of 1. For $m_k$, as it has sensitivity of 1, we can directly obtain the private $m_k$, i.e., $\tilde{m}_k = |\{i : \|g_i^k\| > C\}| + N(0, \sigma_2^2 \mathbb{I})$. In practice, we use $\sigma_2 \approx 10\sigma_1$ which produces a negligible additional cost in the overall privacy budget.

## 5.2 ADAPTIVE NOISE INJECTION STRATEGY

The noise added is $N(0, \sigma^2 C^2 \mathbb{I})$, where the noise multiplier $\sigma$ is controlled by privacy parameters $\epsilon$, $\delta$, and the sampling probability (Mironov et al., 2019; Wei et al., 2022). If the sampling probability is not limited, then it may become too large, and lead to too much noise, which in turn affects the model utility. We defend this obstacle by an adaptive noise injection strategy.

Specifically, we maintain a list of group probabilities, and update each group with its noisy average clipped/normalized gradient norm $\widetilde{\mathcal{G}}_k$ in each epoch by exponentiated gradient ascent as $q'_k \leftarrow q'_k \exp(\eta_q \cdot \widetilde{\mathcal{G}}_k)$, where step size $\eta_q$ can adaptively adjust the group probabilities, so them would not be too large to affect model utility. In addition, we re-normalize the group probabilities to make this process more stable: $q_k \leftarrow q'_k / \sum_{i=1}^{K} q'_i$.

A relatively small group probability $q_k$ can produce a small $\sigma$, eventually resulting in less injected noise, thus improving privacy-fairness trade-off. Adaptive noise could affect the original privacy guarantee, hence we conduct privacy analysis to ensure Algorithm 1 can satisfy the RDP guarantee in Section 5.3.

## 5.3 PRIVACY ANALYSIS OF FDPG

This section establishes the privacy guarantee of FDPG. The sampling probability of FDPG is adaptive which affects the injected noise, hence we first give the upper bound of the sampling probability, and then derive the total privacy loss. Theorem 3 gives the proof of RDP in the training phase, then we use Lemma 1 to convert it to $(\epsilon, \delta) - DP$. The proof can be found in Appendix A.3.

**Theorem 3 (Privacy Loss of FDPG)** *For any $0 \leq \delta \leq 1$, integer $\alpha > 1$, consider FDPG with sampling probability $p \leq p^*$ and noise multiplier $\sigma$, the privacy loss of FDPG satisfes:*

$$(\epsilon, \delta) = (R_{FDPG} + \frac{\log(1/\delta) + (\alpha - 1)\log(1 - 1/\alpha) - \log(\alpha)}{\alpha - 1}, \delta), \qquad (6)$$

*where $R_{FDPG} = \frac{1}{\alpha - 1} \ln \left( \sum_{m=0}^{\alpha} \binom{\alpha}{m} (1 - p^*)^{\alpha - m} (p^*)^m \exp \left( \frac{(m^2 - m)}{2\sigma^2} \right) \right)$.*

## 6 EXPERIMENT

In this section, we conduct experiments to demonstrate the performance of FDPG over five datasets and popular machine learning models. We compare FDPG with four differentially private baselines, namely DP-SGD (Abadi et al., 2016), DPSUR (Fu et al., 2024), AUTO-S (Bu et al., 2024), Disk (Zhang et al., 2025), and three state-of-the-art differentially private and fair baselines, DPSGD-F (Xu et al., 2021), DP-IS-SGD (Kulynych et al., 2022) and DPSGD-Global-Adapt (Esipova et al., 2023). The experiment is conducted over two tabular datasets, including Adult and Dutch (van der

Laan, 2000), and two image datasets: MNIST (LeCun & Cortes, 2010) and CelebA (Liu et al., 2015), and a text dataset the Blog Authorship Corpus (Blog) (Schler et al., 2006).

We use the overall classification accuracy to measure the model utility and group classification accuracy to demonstrate the worse group performance like plenty fairness works. Moreover, to measure private model fairness, we include privacy cost gap as in (Bagdasaryan et al., 2019; Xu et al., 2021; Esipova et al., 2023). Fairness of a private model can be measured in terms of the disparate impact by the cost of adding privacy to a non-private model on the protected groups. Privacy cost for group $a$ represents the reduction in accuracy of group $a$ between the private model and its non-private counterpart, that is $\pi_a = \text{acc}(\theta^*; D_a) - \mathbb{E}_{\tilde{\theta}}[\text{acc}(\tilde{\theta}; D_a)]$. For any group $a, b \in S$, the privacy cost gap is $\pi = \max|\pi_a - \pi_b|$. Smaller $\pi$ means less disparate impact, so the goal of a fair private model is to minimize the privacy cost gap. The best and second-best results are highlighted in **bold** and underline. We report the results of 5 independent trials. For all experiments, full details are provided in App. C.

Table 1: Accuracy and Fairness Metric for Tabular Datasets. For Adult and Dutch, the privacy budget is $\epsilon = 3.4$ and $\epsilon = 2.3$ respectively.

| Method | Adult | | | | Dutch | | | |
| --- | --- | --- | --- | --- | --- | --- | --- | --- |
| | Accuracy | | | Fairness | Accuracy | | | Fairness |
| | Male | Female | Total ↑ | $\pi\downarrow$ | Male | Female | Total ↑ | $\pi\downarrow$ |
| NON-DP | 80.78±0.6 | 92.29±0.3 | 86.54±0.4 | - | 86.93±0.1 | 79.95±0.4 | 83.45±0.2 | - |
| DP-SGD | 70.32±0.8 | 88.61±0.2 | 79.47±0.5 | 6.78±0.8 | 86.54±0.2 | 76.08±0.6 | 81.22±0.3 | 3.48±0.7 |
| DPSUR | 69.07±0.8 | 88.47±0.2 | 78.64±0.6 | 7.89±0.6 | 85.12±0.3 | 73.90±0.8 | 79.53±0.5 | 4.24±0.8 |
| DPSGD-F | 79.19±0.7 | 90.23±0.4 | **84.73±0.5** | 0.46±0.2 | 87.01±0.2 | 76.88±0.6 | 81.88±0.3 | 3.15±0.6 |
| DP-IS-SGD | 69.84±0.8 | 88.56±0.3 | 79.21±0.5 | 7.21±0.4 | 87.02±0.2 | 73.82±0.6 | 80.44±0.4 | 6.22±0.6 |
| DPSGD-G.-A. | 76.89±1.0 | 89.14±0.3 | 83.02±0.7 | 0.73±0.6 | 86.50±0.2 | 78.78±0.2 | 82.61±0.2 | **0.74±0.3** |
| DiSK | 69.67±0.7 | 88.39±0.3 | 79.02±0.5 | 7.21±0.3 | 86.43±0.1 | 76.03±0.5 | 81.12±0.2 | 3.42±0.6 |
| FDPG | 78.28±0.6 | 89.72±0.4 | 84.01±0.5 | **0.07±0.5** | 86.64±0.1 | 78.79±0.2 | **82.73±0.2** | 0.87±0.5 |

Table 2: Accuracy and Fairness Metric for Image Datasets. For MNIST, the privacy budget is $\epsilon = 5.9$ and $\epsilon = 3$ respectively. For CelebA, the privacy budget is $\epsilon = 2.5$ and $\epsilon = 8$ respectively.

| Method | MNIST | | | | CelebA | | | |
| --- | --- | --- | --- | --- | --- | --- | --- | --- |
| | Accuracy | | | Fairness | Accuracy | | | Fairness |
| | class2 | class8 | Total ↑ | $\pi\downarrow$ | Male | Female | Total ↑ | $\pi\downarrow$ |
| NON-DP | 97.79±0.4 | 83.86±2.5 | 96.82±0.3 | - | 95.86±0.1 | 99.09±0.0 | 97.84±0.1 | - |
| DP-SGD | 89.01±0.1 | 25.27±2.2 | 85.28±0.3 | 49.81±3.1 | 87.63±0.2 | 97.51±0.1 | 93.70±0.2 | 6.66±0.2 |
| DPSUR | 87.87±0.3 | 0.45±0.5 | 81.30±0.3 | 73.49±2.4 | 87.22±0.4 | 97.50±0.2 | 93.54±0.0 | 7.05±0.5 |
| DPSGD-F | 89.26±0.2 | 59.05±1.6 | 89.15±0.2 | 16.28±2.3 | 93.68±0.3 | 98.40±0.1 | 96.58±0.1 | 1.50±0.5 |
| DP-IS-SGD | 86.18±0.6 | 87.04±0.3 | **90.38±0.2** | 14.79±2.6 | 81.54±2.7 | 80.44±1.2 | 81.12±2.0 | 4.32±2.3 |
| DPSGD-G.-A. | 88.73±0.2 | 35.78±2.3 | 86.21±0.3 | 39.02±3.2 | 93.76±0.1 | 98.82±0.0 | 96.87±0.0 | 1.83±0.2 |
| DiSK | 89.01±0.1 | 25.30±2.3 | 85.29±0.4 | 49.78±2.3 | 87.58±0.1 | 97.73±0.3 | 93.67±0.3 | 8.60±0.2 |
| FDPG | 89.67±0.4 | 73.90±1.0 | 89.73±0.2 | **1.84±2.7** | 94.32±0.2 | 98.87±0.0 | **97.12±0.1** | **1.32±0.2** |
| NON-DP | 99.03±0.1 | 86.71±2.0 | 97.70±0.2 | - | 96.30±0.2 | 99.28±0.1 | 98.13±0.1 | - |
| AUTO-S | 98.54±0.2 | 56.71±5.2 | 94.22±0.5 | 29.52±4.7 | 93.73±0.2 | 98.83±0.0 | 96.86±0.1 | 2.12±0.4 |
| DP-IS-SGD | 78.79±3.3 | 72.40±6.6 | 81.35±1.0 | **5.92±9.4** | 93.81±0.3 | 98.86±0.0 | 96.91±0.1 | 2.07±0.4 |
| FDPG | 97.82±0.3 | 77.35±5.5 | **95.82±0.6** | 8.16±6.0 | 94.39±0.1 | 98.87±0.1 | **97.10±0.1** | **1.60±0.1** |

## 6.1 MAIN RESULTS

### 6.1.1 SINGLE PROTECTED GROUP ATTRIBUTE SETTING

**Tabular Datasets** Table 1 shows the model accuracy and fairness metric on tabular datasets. We observe that DP-SGD and DPSUR both have a negative impact on the disadvantaged group (male in Adult and female in Dutch) on tabular datasets compared with non-private SGD (NON-DP). Both Adult and Dutch are balanced small scale datasets, hence DPSGD-F, DPSGD-G.-A and our method achieve comparable results on both fairness and utility performance, and DP-IS-SGD fail to effectively achieve fairness in this case. However, FDPG achieves optimal or suboptimal results on both datasets, whereas DPSGD-F and DPSGD-G.-A. only achieve optimal or suboptimal results on one of them.

**Image Datasets** Table 2 shows the accuracy and fairness results on image datasets. We observed that state-of-the-art DP gradient methods DPSUR and AUTO-S both have disparate impact on image datasets. On CelebA dataset, our framework FDPG outperforms other baselines in terms of both accuracy and fairness. On MNIST dataset, FDPG obtains the best

fairness result with slightly lower accuracy (89.73%) than DP-IS-SGD (90.38%) while DP-IS-SGD has a much worse fairness result (14.79) than FDPG (1.84) in gradient clipping setting. In gradient normalization setting, DP-IS-SGD achieves optimal fairness performance. However, this comes at the expense of a significant reduction in accuracy, while FDPG has the best accuracy-fairness trade-off. To further understand the improvement of our framework in terms of fairness, we show the gradient variance of different groups in Fig. 4 (cf. Figs. 8 in App. D.5 for CelebA). FDPG can help reduce the gradient variance of the disadvantaged groups (class 8 and Male), leading to a smaller gradient variance gap among groups.

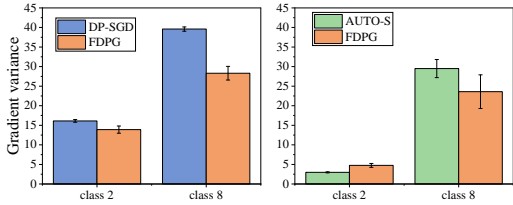

Figure 4: Gradient variance for class 2 and class 8 in MNIST.

### 6.1.2 MULTIPLE PROTECTED GROUP ATTRIBUTES SETTING

We conducted multiple protected group attributes experiments on CelebA and Blog datasets, the results are shown in Table 3. In CelebA, DPSUR performs poorly in terms of accuracy and fairness, because DPSUR requires more iterations to select updates that are beneficial to model performance, poor results may be obtained given relatively small number of epochs. We can also observe DP-IS-SGD achieves much worse fairness in terms of privacy cost gap $\pi$. Although DP-IS-SGD enforces accuracy parity, it significantly upweights minority groups while downweights other groups, where accuracy of minority groups will surpass their non-private counterparts while the majority groups suffer from great performance degradation (more than 10%). Different from accuracy parity, alleviating disparate impact of DP is to restore the performance of each group in DP models close to their non-private level like other fairness-aware baselines do. In both CelebA and Blog datasets, our framework FDPG achieves state-of-the-art accuracy and fairness trade-off. Despite DPSUR has the best accuracy among baselines in Blog, it requires update selection at each iteration, which makes it computationally expensive. In FDPG, accuracy is improved while the accuracy of other fairness-aware baselines is reduced compared with DP-SGD and AUTO-S.

Table 3: Accuracy and Fairness Metric for Text Dataset trained on DistilBERT model under $\epsilon = 5, \delta = 1e - 5$. The best and second-best results are highlighted in **bold** and underline.

| Method | CelebA | | | | | | Blog | | | | | |
|---|---|---|---|---|---|---|---|---|---|---|---|---|
| | Accuracy | | | | | Fairness | Accuracy | | | | | Fairness |
| | group1 | group2 | group3 | group4 | Total ↑ | $\pi$ ↓ | group1 | group2 | group3 | group4 | Total ↑ | $\pi$ ↓ |
| NON-DP | 99.25±0.2 | 95.34±0.9 | 77.32±3.9 | 19.78±4.2 | 93.90±0.1 | - | 73.88±1.1 | 63.52±1.4 | 73.59±1.3 | 66.46±1.3 | 70.51±0.4 | - |
| DP-SGD | 99.00±0.4 | 98.43±0.4 | 30.93±5.3 | 11.33±2.4 | 89.48±0.3 | 49.48±8.7 | 73.06±0.6 | 56.34±0.9 | 73.09±0.8 | 59.58±1.9 | 67.57±0.3 | 7.46±3.0 |
| DPSUR | 99.94±0.1 | 99.94±0.1 | 2.11±3.5 | 1.11±1.9 | 86.89±0.3 | 79.80±7.7 | 73.49±0.4 | 56.13±0.6 | 73.03±0.8 | 56.89±1.7 | **67.62±0.2** | 10.17±2.6 |
| DPSGD-F | 96.78±2.0 | 92.82±3.0 | 78.10±7.0 | 35.33±13.6 | 91.97±1.3 | 21.87±16.5 | 72.03±0.6 | 56.41±1.2 | 72.35±1.1 | 58.14±1.6 | 66.85±0.3 | 7.56±4.0 |
| DP-IS-SGD | 85.86±0.9 | 87.13±0.7 | 91.89±0.6 | 81.89±3.0 | 87.28±0.6 | 75.51±5.6 | 62.87±0.7 | 64.68±0.6 | 71.47±1.5 | 71.05±1.5 | 64.26±0.4 | 15.60±2.4 |
| DPSGD-G.-A. | 98.70±0.4 | 95.86±0.8 | 67.94±4.9 | 19.44±3.9 | 92.78±0.1 | 10.63±4.7 | 67.80±1.0 | 54.39±2.5 | 69.01±1.6 | 56.69±1.7 | 63.45±0.2 | 6.24±2.2 |
| FDPG | 98.60±0.4 | 95.95±0.6 | 71.39±3.2 | 22.67±4.2 | **93.24±0.1** | **9.31±1.6** | 71.76±0.5 | 59.14±0.8 | 72.53±0.8 | 60.35±1.5 | 67.59±0.3 | **5.50±2.4** |
| NON-DP | 99.49±0.2 | 96.39±0.5 | 79.79±1.8 | 28.33±2.8 | 94.89±0.1 | - | 73.88±1.1 | 63.52±1.4 | 73.59±1.3 | 66.46±1.3 | 70.51±0.4 | - |
| AUTO-S | 99.11±0.3 | 96.70±0.6 | 73.04±3.7 | 23.11±4.9 | **94.01±0.1** | 7.78±2.7 | 73.63±0.4 | 55.11±0.4 | 73.06±0.6 | 57.62±0.6 | 67.46±0.2 | 9.75±1.8 |
| DP-IS-SGD | 85.43±2.9 | 85.90±2.1 | 94.07±1.3 | 86.67±3.6 | 86.74±1.9 | 72.40±8.9 | 63.17±0.7 | 64.71±0.8 | 71.44±1.1 | 70.34±1.2 | 64.42±0.2 | 14.58±2.5 |
| FDPG | 98.47±0.4 | 95.98±0.5 | 75.40±2.9 | 26.94±2.9 | 93.75±0.4 | **5.86±2.6** | 72.98±0.3 | 57.25±0.6 | 72.53±0.3 | 59.26±0.2 | **67.74±0.1** | **6.87±1.8** |

## 7 CONCLUSION

In this paper, we explored the relationship between gradient variance and fairness in private model training with DP gradients. Firstly, we conducted plentiful experiments to investigate the impact of gradient variance on model fairness in DP gradients where groups with larger gradient variance suffer from more performance degradation than others. Subsequently, we further verify the relationship through theoretical proofs. Next, we proposed a framework called FDPG to mitigate the disparate impact while maintaining model utility, which consists of a fairness-aware sampling mechanism and an adaptive noise injection strategy. Finally, experimental results on both single and multiple protected group attributes settings confirmed the effectiveness of FDPG.

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

# A   THEORETICAL RESULTS

## A.1   RÉNYI DIFFERENTIAL PRIVACY

To ensure that the privacy budget is not exceeded throughout model training, we use Rényi differential privacy (RDP) (Mironov, 2017) for a better track of the privacy loss, which is a relaxed version of DP. Compared to $(\epsilon, \delta)$-DP, RDP provides an operationally more convenient, quantitatively more accurate way to track the cumulative privacy loss from a composition of multiple mechanisms. RDP is defined based on Rényi divergence as follows.

**Definition 4 (Rényi divergence (van Erven & Harremoës, 2014))** *Given two probability distributions $\mathcal{P}$, $\mathcal{Q}$, $x \in \mathcal{D}$, the Rényi divergence of a finite order $\alpha > 1$ is defined as*

$$D_\alpha(\mathcal{P}\|\mathcal{Q}) \triangleq \frac{1}{\alpha - 1} \ln \int_{\mathcal{X}} q(x) \left(\frac{p(x)}{q(x)}\right)^\alpha dx,$$

*where $p(x)$, and $q(x)$ denotes the density of $\mathcal{P}$ or $\mathcal{Q}$ at $x$ respectively.*

**Definition 5 (Rényi Differential Privacy (Mironov, 2017))** *A randomized mechanism $\mathcal{M} : \mathcal{D} \to \mathbb{R}$ with domain $\mathcal{D}$ and range $\mathbb{R}$ satisfies $(\alpha, R) - RDP$ if*

$$D_\alpha \left(\mathcal{M}(D)\|\mathcal{M}(D')\right) \leq R,$$

*for any neighbor datasets $D, D'$. $R$ is the privacy budget for RDP.*

The following Lemma defines the standard form for converting $(\alpha, R) - RDP$ to $(\epsilon, \delta) - DP$.

**Lemma 1 (Conversion from RDP to DP (Mironov et al., 2019))** *if a randomized mechanism $\mathcal{M} : \mathcal{D} \to \mathbb{R}$ satisfies $(\alpha, R) - RDP$, then it satisfies $(\epsilon, \delta) - DP$ for any $0 \leq \delta \leq 1$, and*

$$\epsilon = R + \frac{\log(1/\delta) + (\alpha - 1)\log(1 - 1/\alpha) - \log(\alpha)}{\alpha - 1}. \tag{7}$$

In model training with RDP, a fundamental building block is the sampled Gaussian mechanism (SGM), which operates by sampling a subset uniformly from a given dataset, applies a function $f$ on the sample set, and injects Gaussian noise according to sensitivity $S_f$.

**Definition 6 (Sampled Gaussian Mechanism (Mironov et al., 2019))** *Given a function $f$ mapping subsets of $D$ to $\mathbb{R}$, with sensitivity $S_f$, SGM with sampling probability $0 < p \leq 1$ and additive Gaussian noise $N(0, S_f^2 \sigma^2 \mathbb{I}^d)$ is*

$$SG_{p,\sigma}(D) \triangleq f(\{\xi : \xi \in D \text{ is sampled with probability } p\}) + \mathcal{N}\left(0, S_f^2 \sigma^2 \mathbb{I}^d\right).$$

### A.2 PROOF OF THEOREM 2

We analyze the relationship between gradient variance and gradient norm for AUTO-S through convergence analysis. Following the standard assumptions in the SGD literature (Koloskova et al., 2023; Zhang et al., 2020), our analysis is built upon bounded variance (Definition 1) and the following assumption:

**Assumption 1 (($L_0$, $L_1$)-Smoothness)** *For all $x, y \in \mathbb{R}^d$, and there exist constants $L_0 > 0$ and $L_1 \geq 0$, function $f$ satisfies ($L_0$, $L_1$)-smoothness, if*

$$\|\nabla f(\mathbf{x}) - \nabla f(\mathbf{y})\| \leq (L_0 + \|\nabla f(\mathbf{x})\| L_1) \|\mathbf{x} - \mathbf{y}\|. \tag{8}$$

From Assumption 1, we can obtain the following lemma.

**Lemma 2** *If $f(x)$ is ($L_0$, $L_1$)-smooth, for $\forall \mathbf{x}, \mathbf{y} \in \mathbb{R}^d$, $\|\mathbf{x} - \mathbf{y}\| \leq \frac{1}{L_1}$, then it also holds that*

$$f(\mathbf{y}) - f(\mathbf{x}) \leq \langle \nabla f(\mathbf{x}_t), \mathbf{y} - \mathbf{x} \rangle + \frac{(L_0 + \|\nabla f(\mathbf{x})\| L_1)}{2} \|\mathbf{x} - \mathbf{y}\|^2.$$

From Lemma 1, we have

$$
\begin{aligned}
&f(\mathbf{x}_{t+1}) - f(\mathbf{x}_t) \\
&\leq -\eta \langle \nabla f(\mathbf{x}_t), \mathbf{g}(\mathbf{x}_t) + \mathbf{z}_t \rangle \\
&+ \frac{\eta^2 (L_0 + \|\nabla f(\mathbf{x}_t)\| L_1)}{2} \|\mathbf{g}(\mathbf{x}_t) + \mathbf{z}_t\|^2.
\end{aligned}
\tag{9}
$$

Here, $\mathbf{g}(\mathbf{x}_t) = \frac{1}{b} \sum_{\xi_i \in B_t} \frac{\nabla f_{\xi_i}(\mathbf{x}_t)}{\|\nabla f_{\xi_i}(\mathbf{x}_t)\| + \gamma}$ is the average of normalized per-sample gradients, where batch size $b = nq$ by uniform sampling with sampling probability $q$. Now taking the expectation on both sides of Eq. 9, and obtain

$$
\begin{aligned}
&\mathbb{E}\left[f(\mathbf{x}_{t+1}) - f(\mathbf{x}_t)\right] \\
&\leq -\eta \mathbb{E}\left[\langle \nabla f(\mathbf{x}_t), \mathbf{g}(\mathbf{x}_t) + \mathbf{z}_t \rangle\right] \\
&+ \mathbb{E}\left[\frac{\eta^2 (L_0 + \|\nabla f(\mathbf{x}_t)\| L_1)}{2} \|\mathbf{g}(\mathbf{x}_t) + \mathbf{z}_t\|^2\right].
\end{aligned}
\tag{10}
$$

Note that the DP noise is of zero mean, i.e., $\mathbb{E}[\mathbf{z}_t] = 0$, and

$$\mathbb{E}[\|\mathbf{g}(\mathbf{x}_t) + \mathbf{z}_t\|^2] = \mathbb{E}[\|\mathbf{g}(\mathbf{x}_t)\|^2] + \mathbb{E}[\|\mathbf{z}_t\|^2],$$

then

$$
\begin{aligned}
&\mathbb{E}[\|\mathbf{g}(\mathbf{x}_t)\|^2] \\
&= \frac{1}{(nq)^2} \mathbb{E}[\| \sum_i \mathbf{1}_i v_i\|^2] \\
&\leq \frac{1}{(nq)^2} \sum_{i \neq j} \mathbb{E}[\mathbf{1}_i \cdot \mathbf{1}_j] \|v_i\| \|v_j\| + \frac{1}{(nq)^2} \sum_{i=1}^{n} \mathbb{E}[\mathbf{1}_i \cdot \mathbf{1}_i] \|v_i\|^2 \\
&< \frac{n(n-1)q^2 + nq}{(nq)^2} < 2,
\end{aligned}
\tag{11}
$$

where $v_i = \frac{\nabla f_{\xi_i}(\mathbf{x}_t)}{\|\nabla f_{\xi_i}(\mathbf{x}_t)\| + \gamma}$, for $i = 1, 2, ..., n$, and $\|v_i\| < 1$, set $\Pr(\mathbf{1}_i = 1) = q$ to indicate whether the $i$-th sample is selected. Therefore,

$$
\begin{aligned}
&\frac{\eta^2 (L_0 + \|\nabla f(x_t)\| L_1)}{2} E[\|\mathbf{g}(\mathbf{x}_t)\|^2 + \|\mathbf{z}_t\|^2] \\
&\qquad\qquad \leq \frac{\eta^2 (L_0 + \|\nabla f(x_t)\| L_1)}{2} (2 + \sigma^2).
\end{aligned}
\tag{12}
$$

Now, we analyze the first term in Eq. 10. For the case that $\|\nabla f(\mathbf{x}_t)\| \geq 6\tau$, we have that

$$\mathbb{E}\big[\mathbf{g}(\mathbf{x}_t)\big] = \mathbb{E}\big[\frac{\nabla f_\xi(\mathbf{x}_t)}{\|\nabla f_\xi(\mathbf{x}_t)\| + \gamma}\big],$$

let $\alpha_\xi = \frac{1}{\|\nabla f_\xi(\mathbf{x}_t)\| + \gamma}$, then

$$\mathbb{E}\big[\langle \nabla f(\mathbf{x}_t), \mathbf{g}(\mathbf{x}_t)\rangle\big] = \mathbb{E}\big[\alpha_\xi \langle \nabla f(\mathbf{x}_t), \nabla f_\xi(\mathbf{x}_t)\rangle\big].$$

Define $\delta = \mathbf{1}\{\|\nabla f_\xi(\mathbf{x}) - \nabla f(\mathbf{x})\| > 3\tau\}$, and take the conditional expectation as follows:

$$\begin{aligned}
&\mathbb{E}\left[\alpha_\xi \langle \nabla f(\mathbf{x}_t), \nabla f_\xi(\mathbf{x}_t)\rangle\right] \\
&\leq p(\delta = 0)\mathbb{E}\left[\alpha_\xi \langle \nabla f(\mathbf{x}_t), \nabla f_\xi(\mathbf{x}_t)\rangle \,|\delta = 0\right] \\
&+ p(\delta = 1)\mathbb{E}\left[\alpha_\xi \langle \nabla f(\mathbf{x}_t), \nabla f_\xi(\mathbf{x}_t)\rangle \,|\delta = 1\right].
\end{aligned} \tag{13}$$

We bound the first term of Eq. 13 and obtain:

$$\begin{aligned}
&\alpha_\xi \langle \nabla f(\mathbf{x}_t), \nabla f_\xi(\mathbf{x}_t)\rangle \\
&= \alpha_\xi \langle \nabla f(\mathbf{x}_t), \nabla f_\xi(\mathbf{x}_t) - \nabla f(\mathbf{x}_t) + \nabla f(\mathbf{x}_t)\rangle \\
&\geq \alpha_\xi \|\nabla f(\mathbf{x}_t)\|^2 - 3\alpha_\xi \|\nabla f(\mathbf{x}_t)\|\tau \\
&\geq \frac{\alpha_\xi}{2} \|\nabla f(\mathbf{x}_t)\|^2,
\end{aligned}$$

then we need to bound $\alpha_\xi$. Note that $\tau < \|\nabla f(\mathbf{x}_t)\|/6$ and $\|\nabla f_\xi(\mathbf{x}_t) - \nabla f(\mathbf{x}_t)\| \leq 3\tau$, so $\|\nabla f_\xi(\mathbf{x}_t)\| \leq 2\|\nabla f(\mathbf{x}_t)\|$, then

$$\alpha_\xi = \frac{1}{\|\nabla f_\xi(\mathbf{x}_t)\| + \gamma} \geq \frac{1}{2\|\nabla f(\mathbf{x}_t)\| + \gamma}.$$

If $\gamma \leq 6\tau \leq \|\nabla f(\mathbf{x}_t)\|$, then $\frac{1}{2\|\nabla f(\mathbf{x}_t)\| + \gamma} \geq \frac{1}{3\|\nabla f(\mathbf{x}_t)\|}$, so

$$\alpha_\xi \geq \frac{1}{3\|\nabla f(\mathbf{x}_t)\|}.$$

Now bounding the second term

$$\begin{aligned}
&\mathbb{E}\left[\alpha_\xi \langle \nabla f(\mathbf{x}_t), \nabla f_\xi(\mathbf{x}_t)\rangle \,|\delta = 1\right] \\
&\geq -\|\nabla f(\mathbf{x})\| \mathbb{E}\left[\|\alpha_\xi \nabla f_\xi(\mathbf{x})\| \,|\delta = 1\right] \\
&\geq -\|\nabla f(\mathbf{x})\|.
\end{aligned}$$

It remains to bound $p(\delta = 1)$. Using Markov inequality, we have that

$$p(\delta = 1) = p(\|\nabla f_\xi(\mathbf{x}) - \nabla f(\mathbf{x})\|^2 > 9\tau^2) \leq 1/9.$$

So, $p(\delta = 0) = 1 - p(\delta = 1) \geq 8/9$. To sum up, we obtain that

$$\begin{aligned}
-\mathbb{E}\big[\langle \nabla f(\mathbf{x}_t), \mathbf{g}(\mathbf{x}_t)\rangle\big] &\leq -\frac{8}{9}\frac{\alpha_\xi}{2}\|\nabla f(\mathbf{x}_t)\|^2 + \frac{1}{9}\|\nabla f(\mathbf{x}_t)\| \\
&\leq -\frac{5}{27}\|\nabla f(\mathbf{x}_t)\|.
\end{aligned} \tag{14}$$

Now the second term in Eq. 10 is bounded in the same way as $\alpha_\xi$, we have

$$\begin{aligned}
\frac{\eta^2(L_0 + \|f(\mathbf{x}_t)\|L_1)}{2}(2 + \sigma^2) \\
\leq \frac{\eta^2(L_0 + \gamma L_1)(2 + \sigma^2)}{2\gamma}\|f(\mathbf{x}_t)\|.
\end{aligned} \tag{15}$$

Plugging Eq. 14 and Eq. 15 into Eq. 10, and obtain

$$\begin{aligned}
\mathbb{E}\left[f(\mathbf{x}_{t+1}) - f(\mathbf{x}_t)\right] &\leq -\frac{5\eta}{27}\|\nabla f(\mathbf{x}_t)\| \\
&+ \frac{\eta^2(L_0 + \gamma L_1)(2 + \sigma^2)}{2\gamma}\|f(\mathbf{x}_t)\|,
\end{aligned}$$

and with $\eta \leq \frac{4\gamma}{27(L_0 + \gamma L_1)(2 + \sigma^2)}$, we obtain:

$$\mathbb{E}\left[f(\mathbf{x}_t) - f(\mathbf{x}_{t+1})\right] \leq -\frac{\eta}{9}\|\nabla f(\mathbf{x}_t)\|. \tag{16}$$

**Final convergence.** If for all $t$ iterations the gradient norm is large ($\|\nabla f(\mathbf{x}_t)\| \geq 6\tau$) and thus Eq. 16 holds for all the iterations. Averaging over $1 \leq t \leq T+1$, we have

$$\frac{1}{T+1}\sum_{t=0}^{T}\|\nabla f(\mathbf{x}_t)\| \leq \mathcal{O}\left(\frac{f(\mathbf{x}_0)-f^*}{\eta T}\right),$$

If for at least one iteration it happens that $\|\nabla f(\mathbf{x}_t)\| \leq 6\tau$, then it simply holds that

$$\min_{t\in[1,T]}\mathbb{E}\|\nabla f(\mathbf{x}_t)\| \leq O\left(\tau\right).$$

combining these two cases, we have

$$\min_{t\in[1,T]}\mathbb{E}\|\nabla f(\mathbf{x}_t)\| \leq \mathbb{E}[\frac{1}{T+1}\sum_{t=0}^{T}\|\nabla f(\mathbf{x}_t)\|]$$
$$\leq \mathcal{O}\left(\tau + \frac{F_0}{\eta T}\right) \tag{17}$$

where $F_0 = f(\mathbf{x}_0) - f^*$.

The proof is completed.

### A.3  PROOF OF THEOREM 3

For simplicity, we only adopt the clipping method but all the following analysis can be generalized to the normalization operation without fundamental difference. First, the estimated mean clipped gradient using the sampled batch $B$ is

$$\tilde{g} = \frac{1}{b}\left(\sum_{i\in B}\overline{g}_i + \mathcal{N}\left(0, \sigma^2 C^2 \mathbb{I}\right)\right), \tag{18}$$

where $\overline{g}_i$ is the gradient after clipping. $\tilde{g}$ follows a mixture Gaussian distribution:

$$f(D) = \frac{1}{b}\sum_{B}p(B)\mathcal{N}\left(\sum_{i\in B}g_i, \sigma^2 C^2 \mathbb{I}\right), \tag{19}$$

where $p(B)$ stands for the probability of sampling batch from dataset $D$, i.e.,

$$p(B) = \prod_{x_i}^{B}p_i\prod_{x_j}^{D\setminus B}(1-p_j).$$

Next, consider a neighbor dataset $D' = D \cup \{z\}$. The sets of gradients of $D$ and $D'$ are $\{g_i\}_i^N$ and $\{g_i\}_i^N \cup \{g_z\}$, respectively. Given $D'$ is the training set, the distribution of $\tilde{g}$ is

$$f(D') = \frac{1}{b}\sum_{B}p(B)\left((1-p_z)\mathcal{N}\left(\sum_{i\in B}g_i, \sigma^2 C^2 \mathbb{I}\right)+\right.$$
$$\left. p_z\mathcal{N}\left(\sum_{i\in B}g_i + g_z, \sigma^2 C^2 \mathbb{I}\right)\right).$$

We proceed to bound the following Rényi divergence:

$$D_\alpha\Big(f(D') \parallel f(D)\Big) \leq D_\alpha\Big(b\cdot f(D') \parallel b\cdot f(D)\Big)$$

$$\leq \sup_B D_\alpha\left((1-p_z)\mathcal{N}\left(\sum_{i\in B}g_i, \sigma^2 C^2 \mathbb{I}\right)\right.$$

$$\left. + p_z\mathcal{N}\left(\sum_{i\in B}g_i + g_z, \sigma^2 C^2 \mathbb{I}\right) \Big\| \mathcal{N}\left(\sum_{i\in B}g_i, \sigma^2 C^2 \mathbb{I}\right)\right)$$

$$\leq \sup_{\|g_z\|\leq C} D_\alpha\left((1-p_z)\mathcal{N}(0, \sigma^2 C^2 \mathbb{I})\right.$$

$$\left. + p_z\mathcal{N}\left(g_z, \sigma^2 C^2 \mathbb{I}\right) \Big\| \mathcal{N}\left(\mathbf{0}, \sigma^2 C^2 \mathbb{I}\right)\right).$$

where first inequality follows from the data processing inequality for Rényi divergence, the second inequality follows from the joint quasi-convex property of Rényi divergence (van Erven & Harremoës, 2014), and the last inequality follows from the translation and rotation invariance for Rényi divergence (Mironov, 2017). Since Rényi divergence is additive, for any gradient $g_z$ and its corresponding $p_z$, we have

$$D_\alpha \left( (1 - p_z)\mathcal{N}(\mathbf{0}, \sigma^2 C^2 \mathbb{I}) + p_z \mathcal{N}\left(g_z, \sigma^2 C^2 \mathbb{I}\right) \parallel \mathcal{N}\left(\mathbf{0}, \sigma^2 C^2 \mathbb{I}\right) \right)$$

$$= D_\alpha \left( (1 - p_z)\mathcal{N}\left(0, \sigma^2 C^2\right) + p_z \mathcal{N}\left(\|g_z\|, \sigma^2 C^2\right) \parallel \mathcal{N}\left(0, \sigma^2 C^2\right) \right)$$

$$= \frac{1}{\alpha - 1} \ln \left( \sum_{m=0}^{\alpha} \binom{\alpha}{m} (1 - p_z)^{\alpha - m} p_z^m \exp \left( \frac{(m^2 - m)\|g_z\|^2}{\sigma^2 C^2} \right) \right).$$

With $\|g_z\| \leq C$, for DP-SGD, $p_z = \frac{1}{n}$. However, for FDPG, $p_z$ is not fixed, we let $p_z \leq p^*$, $p^*$ is easy to obtain by making it the maximum sampling probability of n samples, i.e., $p^* = max(p_1, ..., p_n)$, we have

$$\sup_{\|g_z\| \leq C} \frac{1}{\alpha - 1} \ln \left( \sum_{m=0}^{\alpha} \binom{\alpha}{m} (1 - p_z)^{\alpha - m} p_z^m \exp \left( \frac{(m^2 - m)\|g_z\|^2}{\sigma^2 C^2} \right) \right)$$

$$\leq \frac{1}{\alpha - 1} \ln \left( \sum_{m=0}^{\alpha} \binom{\alpha}{m} (1 - p^*)^{\alpha - m} (p^*)^m \exp \left( \frac{(m^2 - m)}{\sigma^2} \right) \right)$$

$$\triangleq R_{FDPG}.$$

Finally, we use Lemma 1 to convert it to $(\epsilon, \delta) - DP$. The proof is completed.

## B   ALGORITHM

Algorithm 1 summarizes the main steps of the proposed method.

---

**Algorithm 1** FDPG

---

**Input**: Dataset $D$ with $n$ samples, loss function $f(\mathbf{x})$, batch size $b$, epochs $E$, learning rate $\eta$, clipping threshold $C$, step size $\eta_q$, auxiliary bound $\beta$, noise multipliers $\sigma_1, \sigma_2$
**Output**: $\mathbf{x}_T$, privacy cost $(\epsilon, \delta)$

1: Initialize all group probabilities with 1.
2: **for** epoch $e \in [1, E]$ **do**
3:     **for** iteration $t \in [1, T]$ **do**
4:         Sample batch $B_t$ with probability $p(\xi) = bp(\xi)$
5:         **for** each sample $\xi_i \in B_t$ **do**
6:             Compute per-sample gradient $g_i = \nabla f_{\xi_i}(\mathbf{x}_t)$
7:             $\overline{g}_i = \mathbf{Clip/Normalize}(g_i)$
8:         **end for**
9:         $\tilde{g} = \frac{1}{b} \left( \sum_{i \in B_t} \overline{g}_i + N\left(\sigma_1^2 C^2 \mathbb{I}\right) \right)$
10:         $\mathbf{x}_{t+1} = \mathbf{x}_t - \eta_t \tilde{g}$
11:     **end for**
12:     **for** each group $k \in [K]$ **do**
13:         $q'_k = q_k^t \cdot \exp(\eta_q \cdot \tilde{\mathcal{G}}_k)$
14:         $q_k^{t+1} = q'_k / \sum_{k \in [K]} q'_k$
15:     **end for**
16:     Assign new group probabilities to $n$ samples to get the sampling probability distribution $p(\xi)$
17: **end for**
18: **return** $\mathbf{x}_T$ and $(\epsilon, \delta)$

---

## C  EXPERIMENTAL DETAILS

### C.1  BASELINES

The details of the baselines in our experiments are as follows:

- DP-SGD (Abadi et al., 2016): It is a standard method for private model training with gradient clipping and Gaussian noise.
- DPSUR (Fu et al., 2024): It is a state-of-the-art DP gradient method, which selects updates that lead to convergence and applies those updates to model.
- AUTO-S (Bu et al., 2024): Utilizing gradient normalization to improve model performance and convergence without the need to tune clipping threshold.
- DPSGD-F (Xu et al., 2021): It adaptively adjusts the clipping threshold for each group to compensate for the different utility loss on groups due to clipping.
- DP-IS-SGD (Kulynych et al., 2022): Specifically designed to improve the accuracy of the worse-performing groups while retaining DP guarantees.
- DPSGD-Global-Adapt (DPSGD-G.-A.) (Esipova et al., 2023): It aims to mitigate the disparate impact by scaling gradients by introducing a strict upper bound $Z$ and adaptively updating $Z$ to upper-bound the max gradient norm in a batch.

When compared to DP-SGD and DPSUR, we implement gradient clipping for all methods. When compared to AUTO-S, among the fair baselines, DP-IS-SGD can be implemented in AUTO-S framework by replace gradient clipping with gradient normalization. However, DPSGD-F and DPSGD-G.-A. ensure fairness by modifying the clipping procedure which can not be replaced by gradient normalization. Hence, to demostrate the fairness effect of FDPG in gradient normalization setting, we compare FDPG with DP-IS-SGD.

### C.2  DATASETS AND MODELS

The details of the datasets and models are as follows:

**Adult** consists of 48,842 samples. As is typical in the fairness literature, we use "sex" as the protected group attribute. The classification label is "income" (whether or not income exceeds $\$50,000$). Prior to sampling, the Adult dataset is unbalanced with respect to sex with 30,527 males and 14,695 females. We sample a balanced dataset as in (Xu et al., 2021; Esipova et al., 2023) with 14,000 females and 14,000 males on average.

**Dutch** is preprocessed as (Esipova et al., 2023), for a total of 60,420 samples. We consider "sex" as the protected group attribute. The processed dataset is balanced with respect to "sex" with 30,147 male samples and 30,273 female samples.

**MNIST** contains 60,000 training samples and 10,000 testing samples of handwritten digits, divided into 10 classes with 7,000 grayscale images per category. We choose class 8 as an artificially underrepresented group to compare with the typical class 2 as in fairness-aware literature (Xu et al., 2021; Esipova et al., 2023).

**CelebA** consists of 202,599 face images of various celebrities, along with binary attributes describing each image. We use the binary attribute "Eyeglasses" for the target, the attribute "Male" is our protected group (118,165 females and 84,434 males) in single protected attribute setting. For multiple protected attributes, we use (Male, Blonde) as (Kulynych et al., 2022).

**Blog** contains weblogs written on 19 different topics, collected from the Internet before August 2004 with a total of 681,288 posts and over 140 million words. We binarize age, distinguishing between young ($\leq 35$) and older ($> 35$) authors, and take (gender, age) as protected group attributes, resulting in four different group combinations. We chose two topics of roughly equal size (Technology and Arts), reducing the topic classification task to a binary classification task.

As in (Esipova et al., 2023), we apply the same convolutional neural network (CNN) architecture to two image datasets, i.e., MNIST and CelebA, logistic regression model for Adult, and MLP with two hidden layers of 256 units for Dutch. When compared to AUTO-S, we adopt the same 4-layer

CNN architecture for MNIST and ResNet9 for CelebA as in (Bu et al., 2024). Additionally, we used a pretrained DistilBERT model (66M) for the Blog dataset.

### C.3 PARAMETER SETTINGS

We follow the settings of previous works (Esipova et al., 2023; Bu et al., 2024; Hansen et al., 2024). For comparison with DP-SGD, we set learning rate $\eta = 0.01$, noise multiplier $\sigma = 1$, clipping threshold $C = 0.5$ for Adult, $\eta = 0.8$, $\sigma = 1$, $C = 0.1$ for Dutch, and $\eta = 0.01$, $\sigma = 0.8$, $C = 1$ for MNIST. In CelebA, $\eta = 0.1$ for fairness-aware baselines, $\sigma = 0.8$, $C = 1$. With this, training 20 epochs for tabular datasets, 60 epochs for MNIST and 30 epochs for CelebA with fixed $\delta = 10^{-6}$, which gives $\epsilon = 3.4$ for Adult, $\epsilon = 2.3$ for Dutch, $\epsilon = 5.9$ for MNIST, and $\epsilon = 2.5$ for CelebA. These privacy budget are the same as (Esipova et al., 2023). For comparison with AUTO-S, we adopt the same hyperparameters from (Bu et al., 2024), where $\epsilon = 3$ for MNIST and $\epsilon = 8$ for CelebA. For Blog dataset, We use Adam optimizer, and set $\eta = 0.0005$, $\sigma = 0.8$, $C = 1.2$ and 20 epochs, 16 batch size for comparison with DP-SGD while $C = 1$ for comparison with AUTO-S. In addition, DPSUR selects update during training. To make DPSUR obtain the same privacy as other baselines, we set the noise multiplier $\sigma = 0.86$ for Adult, $\sigma = 0.88$ for dutch, $\sigma = 0.7$ for MNIST, $\sigma = 0.74$ for CelebA, and $\sigma = 0.49$ for Blog in DPSUR. All experiments are implemented on a single RTX3090 (24 GB) NVIDIA GPU with Pytorch framework.

## D ADDITIONAL EXPERIMENTAL RESULTS

### D.1 NORMALIZATION ERROR UNDER DIFFERENT $\gamma$ VALUES

$F(a)$ is visualized in Fig. 5, it is clear that when $a \leq 1 - \gamma$, $F(a)$ has an upper bound.

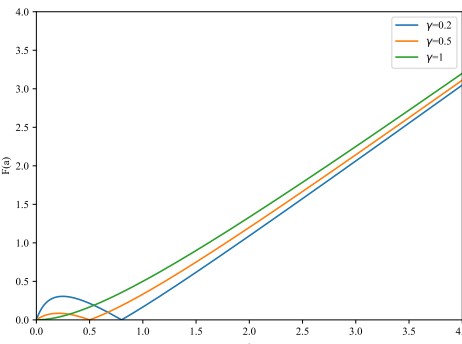

Figure 5: Visualization of normalization error at different $\gamma$.

### D.2 COMPUTATIONAL TIME

In Fig. 6, we visualize the training time required by FDPG and baselines on large scale datasets. FDPG only incurs a negligible overhead compared to DP-SGD and AUTO-S, and is scalable as other private and fair baselines in terms of runtime. In contrast, DPSUR is on average two times as expensive compared to DP-SGD for selecting updates at each iteration.

### D.3 ACCURACY-FAIRNESS TRADE-OFF UNDER DIFFERENT PRIVACY LEVEL

We explore the effect of private and fair methods across different privacy budget $\epsilon$, we choose $\epsilon \in \{1, 3, 5, 7, 9, 10\}$. We use CelebA dataset with group = Male and Blog dataset with group = (gender, age). All other hyperparameters remain unchanged and all methods share the same seeds. The results can be found in Fig. 7.

Observations indicate that FDPG outperforms other baselines in fairness and utility performance on Blog dataset. On CelebA dataset, FDPG outperforms in fairness on strong privacy level. Although

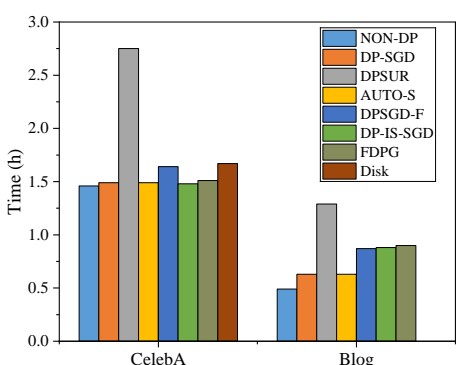

Figure 6: Training time. Training time (in hours) across CelebA and Blog datasets.

DPSGD-F shows better fairness on relatively higher privacy budgets, its sensitivity to privacy budget suggests reduced robustness. Both our method and DPSGD-G.-A. show comparable utility, but our approach outperforms in terms of fairness and more robust to different privacy budget. DP-IS-SGD demonstrates high sensitivity to privacy budget on both datasets. In general, our method achieves an optimal balance between fairness and accuracy.

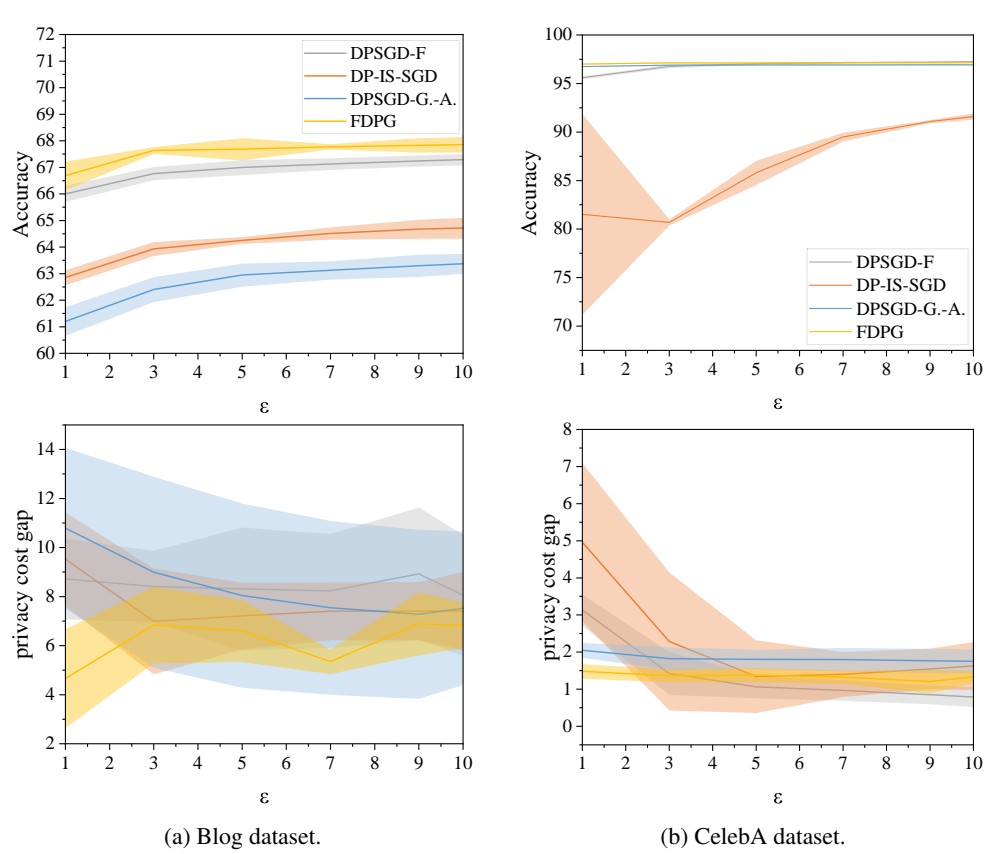

(a) Blog dataset.          (b) CelebA dataset.

Figure 7: Accuracy, privacy and fairness trade-offs for different datasets.

### D.4 HANDLING NEW GROUP FAIRNESS NOTIONS

We consider the Scalability of FDPG to address additional fairness notions. This characteristic arises from the flexibility of the fairness-aware sampling mechanism. We consider standard fairness metrics including Demographic Parity (Feldman et al., 2015), Equal Opportunity (Hardt et al., 2016), and Equalized Odds (Hardt et al., 2016).

Table 4 verifies the fairness metrics obtained by FDPG and the baselines DP-SGD, AUTO-S and NON-DP method. The smaller fairness metrics the lower the fairness violations. Remarkably, the fairness violations reported by FDPG are often significantly lower than those reported by DP-SGD, AUTO-S and NON-DP.

Table 4: Comparison with other fairness notions on Adult dataset.

| Method | Demographic Parity | Equal Opportunity | Equalized Odds |
|--------|--------------------|--------------------|----------------|
| NON-DP | 0.1944 | 0.2079 | 0.2924 |
| DP-SGD | 0.1343 | 0.2292 | 0.2748 |
| AUTO-S | 0.1059 | 0.1948 | 0.2250 |
| FDPG | **0.0530** | **0.0949** | **0.1083** |

### D.5 GRADIENT VARIANCE OF EACH GROUP IN IMAGE DATASETS

Fig. 8 shows the gradient variance of different groups on the CelebA dataset. The results indicate that FDPG reduces the gradient variance of disadvantaged groups (e.g., Male), thereby decreasing the gradient variance gap among groups and improving fairness.

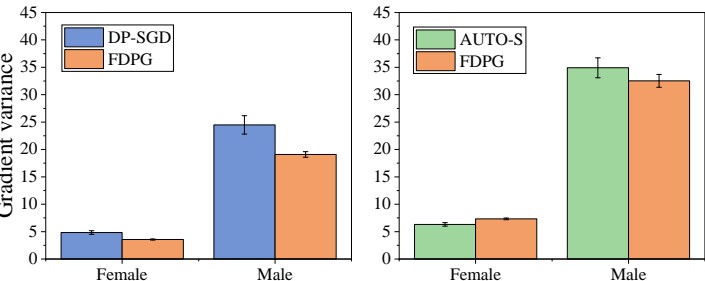

Figure 8: Gradient variance for Female and Male in CelebA.

