# OpenReview forum: "Private and debiased model training: A fair differential privacy gradient framework"
_ICLR.cc/2026/Conference — ICLR 2026 Conference Withdrawn Submission_

### Official Review · Reviewer_HD1f · 2025-10-27

**Soundness:** 3
**Presentation:** 2
**Contribution:** 2
**Rating:** 4
**Confidence:** 3

**Summary:**

This paper discusses the fairness degradation when models are trained with differential privacy. The authors observe that adding DP noise or clipping causes uneven performance loss across demographic groups, leading to biased model behavior. Through empirical analysis and theoretical derivation, the paper reveal the interdependency between gradient variance and model fairness, showing that groups with larger gradient variance tend to experience greater accuracy degradation under DP training. To mitigate this issue, the authors propose a framework called Fair Differential Privacy Gradient (FDPG), which integrates two main components:Fairness-aware sampling mechanism and Adaptive noise injection strategy. Experiments demonstrate that FDPG reduces fairness disparities while maintaining comparable accuracy relative to privacy-focused and fairness-focused methods.

**Strengths:**

1. The paper theoretically analyzes the relationship between gradient variance and model fairness under differential privacy. The proposed FDPG framework integrates fairness-aware sampling with adaptive noise control, forming a new composite strategy.
2. The paper provides both theoretical proofs and empirical validation across multiple datasets and baselines. The theoretical analysis is logically consistent, providing mathematical reasoning that interprets empirical findings. The experimental evaluation is extensive, covering multiple modalities and comparing against both privacy-focused and fairness-focused baselines.
3. The paper’s motivations are clearly stated, with well-structured problem formulation and detailed algorithm descriptions. The authors convey the key intuition that reducing gradient variance across groups improves fairness.

**Weaknesses:**

1. The proposed FDPG framework combines existing ideas (i.e., sampling reweighting and adaptive noise scaling) rather than introducing a fundamentally new algorithmic principle. The novelty lies more in the application.
2. While the paper links gradient variance to fairness degradation, this theoretical insight builds on prior empirical observations. The overall improvement over existing DP–fairness methods is incremental.
3. The experiments lack ablation studies that isolate the contributions of the sampling and noise injection components. It remains unclear which part of the framework drives the observed fairness improvements.
4. Lack of scalability and efficiency discussion: The method’s computational cost and applicability to large-scale models are not discussed. This limits the practical relevance and generalization of the approach.

**Questions:**

- Could you elaborate on what is fundamentally new about FDPG compared to prior fairness-aware DP methods such as DPSGD-F (Xu et al., 2021), DP-IS-SGD (Kulynych et al., 2022), or DPSGD-Global-Adapt (Esipova et al., 2023)? Specifically, how does your fairness-aware sampling differ mathematically from existing adaptive clipping or gradient alignment strategies that also rebalance group effects?
- What is the impact of the fairness-aware sampling mechanism and the adaptive noise injection strategy? Which component contributes more to fairness improvement?
- Could you provide more intuition or visualization explaining why disadvantaged groups have higher gradient variance under DP-SGD or Auto-S？Does this relationship hold consistently across architectures and data distributions?

---

### Official Review · Reviewer_V5qV · 2025-10-29

**Soundness:** 3
**Presentation:** 1
**Contribution:** 3
**Rating:** 6
**Confidence:** 4

**Summary:**

The paper looks at fairness-aware private training of models across multiple modalities. The authors point out  (empirically with accompanying theoretical results) that  high gradient variance in a group is correlated with a high gradient norm for that group (especially for protected/minority groups), leading to disparate impacts when using differentially private training methods like DP-SGD that utilize gradient clipping. To that end, they propose a fairness-aware private training procedure that oversamples from groups with high gradient variance to produce more fair private trained models.

**Strengths:**

**[S1]** Provides a good discussion of the preliminaries/technical background for newcomers.

**[S2]** Provides valuable and interesting insights on using gradient variance as a measure of vulnerability of a group to disparate impacts/unfairness and on how gradient norm can be reduced by reducing the grad variance.

**[S3]** Good breadth of baselines (including SoTA fair DPSGD methods) and datasets, and good cross-modality selections of datasets provides a comprehensive evaluation.

**[S4]** Good in-depth comparison with baselines, touching upon computational expense, accuracy tradeoffs across groups, etc.

**[S5]** The provided theoretical results concretely corroborate the insights provided by the authors on how gradient variance (and other quantities) influence(s) gradient norms for both DP-SGD and AUTO-S

**[S6]** Very well motivated sampling method based on gradient variance. However this strength is stated with a caveat about its claimed novelty, refer to weakness W2.

**Weaknesses:**

Despite the stated strengths, this work suffers from some non-trivial weaknesses that may impact its soundness and contribution.

**[W1]** All the figures and tables in the paper are very poorly presented, and certainly not up to par for a conference paper at such a venue. In particular, they are all very small and very hard to read and interpret. Any recommendation for acceptance from my end will **strongly** hinge on the authors agreeing to improve the presentation of their results (making figures bigger, making sure text in figures, captions, tables, etc. is of a comparable size to the main text, etc.). While this might seem like a relatively peripheral concern, the fact that all the tables and figures have been presented in such a manner is inappropriate for a camera ready version, and I cannot (at least strongly) support a paper in this form.

**[W2]** The authors make an incorrect claim that fairness-aware sampling methods do not exist (refer to [1]). The mentioned paper [1] does fair sampling, including in more involved settings like the presence of DP noise in the priors and mentions how oversampling protected groups may improve fairness for them. While the proposed sampling strategy by the authors makes sense for their paper and is sound, their method does not constitute the first fairness-aware data sampling method and the text must acknowledge this to avoid misrepresentation of novelty.

**[W3]** The reported values of $\varepsilon$ are ambiguous; it is not clearly reported how the reported empirical values arise from Theorem 3. Refer to question Q1.

**[W4]** Minor weakness: there are typographical errors in the paper, for example, on Page 2 in the second bullet point of the contribution uses “debaised” instead of “debiased”

---

## References

[1] Ko, J., Ziani, J., Das, S., Williams, M., & Fioretto, F. (2025). Fairness Issues and Mitigations in (Differentially Private) Socio-Demographic Data Processes. Proceedings of the AAAI Conference on Artificial Intelligence, 39(27), 28160-28167. https://doi.org/10.1609/aaai.v39i27.35035

**Questions:**

**[Q1]** Are the reported values of $\varepsilon$ in line with Theorem 3? If so, please report the calculations/parameter values for that calculation somewhere, especially the sampling probability.

**[Q2]** Can you please address my concerns in the weaknesses section, particularly W1 and W2? This is important to address the key concerns I have that may prevent me from strongly championing this work.

---

### Official Review · Reviewer_qyY2 · 2025-10-31

**Soundness:** 2
**Presentation:** 2
**Contribution:** 2
**Rating:** 2
**Confidence:** 4

**Summary:**

This paper claims that the fairness gap in DP-trained models is caused by higher gradient variance in disadvantaged groups. It proposes FDPG, a framework that uses fairness-aware sampling (oversampling high-variance groups) to reduce this gap, claiming better utility and fairness than existing methods.

**Strengths:**

1. Focus on the important problem
2. The proposed method seems to be interesting.
3. Compare with multiple methods

**Weaknesses:**

1. Baseline: I think it is good to compare with different baselines, but I do not think you picked the right one. Some baseline focuses on improving the efficiency of DP training, such as Auto-S and DPSUR. Why do you need to compare with those methods? I think you need to compare with a method which is showing a better privacy utility trade-off off such as De, Soham, et al. "Unlocking high-accuracy differentially private image classification through scale." arXiv preprint arXiv:2204.13650 (2022). It is better to have more recent baselines that focus on improving the fairness of DP training.
2. Datasets: You are using some toy datasets. Please consider using a more complex one.
3. Experiments: Why only report one or two epsilons for one dataset? Multiple epsilons are needed to show the proposed method's performance.

**Questions:**

1. Can you provide a formal proof linking gradient variance directly to the "privacy cost gap" metric, rather than just standard convergence bounds?
2. Justify the use of last-epoch gradient norms as a stable proxy for current-epoch gradient variance.
3. Where is the formal privacy accounting for the per-epoch release of group statistics?

---

### Official Review · Reviewer_gu9N · 2025-11-04

**Soundness:** 2
**Presentation:** 2
**Contribution:** 2
**Rating:** 4
**Confidence:** 3

**Summary:**

The paper investigates the relationship between gradient variance and model fairness. The authors provide empirical evidence showing that groups with larger gradient variance tend to suffer from higher loss, supported by theoretical justification. In addition, they propose a new fair differentially private gradient method with adaptive sampling to mitigate performance disparities across groups.

**Strengths:**

The authors provide a new perfective on studying the model fairness under differential privacy. The idea of improving model fairness by controlling gradient variance looks interesting.

**Weaknesses:**

1. Theorem 1 is not sufficiently convincing in establishing the correlation between gradient variance and group loss. The theorem demonstrates that a smaller $\tau$ leads to a smaller gradient norm when training is performed on a single distribution. However, in Section 4.1, both training and sampling are conducted on a mixture of multiple group distributions. The paper does not clearly explain how the conclusion drawn from the single-distribution case generalizes to the multi-distribution setting. This connection appears far-fetched and requires a more rigorous justification.
2. t is surprising that the upper bound in Theorem 2 appears to be independent of $\gamma$. Does this imply that the result still holds as $\gamma \to \infty$, or does the theorem require setting $\gamma$ within a specific range for the bound to be valid?
3. Theorem 3 is difficult to interpret in its current form. It would be helpful if the authors could provide a simplified version of the bound using Big-O notation. This would make it easier to understand the dependence of the result on key parameters such as $m$ and $\alpha$.

Minor Comments:
1. In Equations (2) and (4), the notation $\lVert \cdot \rVert$ should be used instead of $|\cdot|$ when bounding $\bar{G}_k - G_k$, since $G_k$ represents a vector.
2. Some notations, such as $b^k$, can be easily confused with exponentiation. It would be clearer to use a subscript notation, e.g., $b_k$.

**Questions:**

Questions are included in "Weaknesses" section.

---

### Official Review · Reviewer_uoZN · 2025-11-08

**Soundness:** 3
**Presentation:** 3
**Contribution:** 3
**Rating:** 4
**Confidence:** 4

**Summary:**

The paper studies disparate impact induced by differentially private (DP) gradient training, and proposes FDPG, a training framework that aims to mitigate group-wise fairness degradation while preserving utility and DP guarantees. The contributions include: indicate the relations between gradient variance and fairness in DP gradients, and present theoretical proof that gradient variance canbe utilized to reduce disparate impact, thus enhancing model fairness; a fairness-aware sampling mechanism by reweighting group samples in a privacy-preserving way and develop an adaptive noise injection strategy to control the influence of noise; a novel Fair Differential Privacy Gradient framework for private and debaised model training. Experiments across tabular and image datasets show improved fairness with competitive accuracy vs. several baselines.

**Strengths:**

1.	The paper addresses a well-known issue in DP and fairness area, that DP training disproportionately degrades minority/disadvantaged groups, and the author give a theoretic analysis and indicate the relations between gradient variance and fairness in DP gradients.
2.	It give an empirical and theoretical linkage, that disadvantaged groups exhibit higher gradient variance is supported by convergence-style analyses, and the  variance controls residual optimization error under both clipping (DP-SGD) and normalization (AUTO-S).
3.	The paper proposes a general framework, so called FDPG, which combines group-aware sampling probabilities estimated via privatized group statistics and adaptive control of sampling through exponentiated updates, satisfying the RDP guarantee.
4.	It presents comprehensive experiments, on Multiple datasets, with both single and multiple protected attributes, and comparisons against several DP-fairness baselines.  The presentation and writing is very clear.

**Weaknesses:**

1.	The performance is competitive but not strong enough. In Tab. 1, 2, 3, the total accuracy and fairness scores are not the best.
2.	The DP and fairness problem has been discussed a lot in the community. The key algorithmic novelty of this paper is a fairness-aware sampling probability updated by privatized group gradient norms and exponentiated updates. This is sensible but somewhat incremental conceptually.
3.	The variances in the result tables seems too large.
4.	The sensitive to the group size and imbalance, should be discussed. The proposed fairness-aware sampling mechanism is mainly for fairness via gradient variance, thus a more controlled design where imbalance is varied, would strengthen the variance mechanism.
5.	The proposed sampling mechanism introduces additional affect to oringinal data distribution, the robustness analysis will make this work more solid.

**Questions:**

1.	Further improvement and experiments might be needed to increase practical performance. Although FDPG achieves first or second performances, the scores don’t have a clear advantage, considering limit trials and performance variances.
2.	The authors should give more explanation about the fairness score pi, that as the mean and std of accuracy differences, it seem too large for many methods.
3.	How do you set β in practice? Is it a fixed global norm cap, per-layer, or adaptive? How sensitive are results to β, and how is its privacy impact accounted for precisely?
4.	Can you report an ablation where fairness-aware sampling is driven by per-group variance estimates directly (with privatization), instead of gradient norms, to quantify the gap vs. your surrogate?

---

### Note · Authors · 2025-11-13

I have read and agree with the venue's withdrawal policy on behalf of myself and my co-authors.